# Proteomic Analysis Identifies NDUFS1 and ATP5O as Novel Markers for Survival Outcome in Prostate Cancer

**DOI:** 10.3390/cancers13236036

**Published:** 2021-11-30

**Authors:** Robert Wiebringhaus, Matteo Pecoraro, Heidi A. Neubauer, Karolína Trachtová, Bettina Trimmel, Maritta Wieselberg, Jan Pencik, Gerda Egger, Christoph Krall, Richard Moriggl, Matthias Mann, Brigitte Hantusch, Lukas Kenner

**Affiliations:** 1Department of Pathology, Medical University of Vienna, 1090 Vienna, Austria; bettina.trimmel@meduniwien.ac.at (B.T.); maritta.wieselberg@meduniwien.ac.at (M.W.); Jan.Pencik@lbicr.lbg.ac.at (J.P.); gerda.egger@meduniwien.ac.at (G.E.); brigitte.hantusch@meduniwien.ac.at (B.H.); 2Department of Otolaryngology, University Hospital, LMU Munich, 81377 Munich, Germany; 3Institute for Research in Biomedicine, Università della Svizzera Italiana, 6500 Bellinzona, Switzerland; matteo.pecoraro@irb.usi.ch; 4Institute of Animal Breeding and Genetics, University of Veterinary Medicine Vienna, 1210 Vienna, Austria; Heidi.Neubauer@vetmeduni.ac.at (H.A.N.); richard.moriggl@lbicr.lbg.ac.at (R.M.); 5Central European Institute of Technology, Masaryk University, 60177 Brno, Czech Republic; k.trachtova@gmail.com; 6Christian Doppler Laboratory for Applied Metabolomics (CDL-AM), Medical University of Vienna, 1090 Vienna, Austria; 7Division of Nuclear Medicine, Department of Biomedical Imaging and Image-Guided Therapy, Medical University of Vienna, 1090 Vienna, Austria; 8Ludwig Boltzmann Institute Applied Diagnostics, 1090 Vienna, Austria; 9Institute for Statistics, Medical University of Vienna, 1090 Vienna, Austria; christoph.krall@univie.ac.at; 10Department of Proteomics and Signal Transduction, Max Planck Institute of Biochemistry, 82152 Martinsried, Germany; mmann@biochem.mpg.de; 11Center for Biomarker Research in Medicine (CBmed), 8010 Graz, Austria; 12Unit for Laboratory Animal Pathology, University of Veterinary Medicine Vienna, 1210 Vienna, Austria

**Keywords:** prostate cancer, FFPE-proteomics, OXPHOS, NDUFS1, ATP5O, STAT3, transcriptomics

## Abstract

**Simple Summary:**

Due to the heterogeneity of prostate cancer (PCa), it is still difficult to provide risk stratification. Metabolic changes in PCa tissue have been described during tumor progression at genetic and transcriptomic level, but these have not yet clearly contributed to improved diagnosis and therapy. The aim of our study was to identify novel markers for aggressive prostate cancer in a proteomics-derived dataset by immunohistochemical analysis and correlation with transcriptomic data. Here, we provide potential new markers—NDUFS1 and ATP5O—for risk stratification in PCa. Additionally, we reveal for the first time a concordant increase of NDUFS1/ATP5O of mRNA expression in transcriptomic datasets and at protein level.

**Abstract:**

We aimed to identify novel markers for aggressive prostate cancer in a STAT3-low proteomics-derived dataset of mitochondrial proteins by immunohistochemical analysis and correlation with transcriptomic data and biochemical recurrence in a STAT3 independent PCa cohort. Formalin-fixed paraffin-embedded tissue (FFPE) sample selection for proteomic analysis and tissue-microarray (TMA) generation was conducted from a cohort of PCa patients. Retrospective data analysis was performed with the same cohort. 153 proteins differentially expressed between STAT3-low and STAT3-high samples were identified. Out of these, 46 proteins were associated with mitochondrial processes including oxidative phosphorylation (OXPHOS), and 45 proteins were upregulated, including NDUFS1/ATP5O. In a STAT3 independent PCa cohort, high expression of NDUFS1/ATP5O was confirmed by immunocytochemistry (IHC) and was significantly associated with earlier biochemical recurrence (BCR). mRNA expression levels for these two genes were significantly higher in intra-epithelial neoplasia and in PCa compared to benign prostate glands. NDUFS1/ATP5O levels are increased both at the mRNA and protein level in aggressive PCa. Our results provide evidence that NDUFS1/ATP5O could be used to identify high-risk PCa patients.

## 1. Introduction

Prostate cancer (PCa) causes about 3.8% of all cancer related deaths in men, making it the second most frequent malignancy and fifth leading cause of death in men in western countries [1]. Diagnosis is conducted via evaluating prostate-specific antigen (PSA) levels and by digital rectal examination, following prostate needle biopsy for histological evaluation. The disease is stratified using the Gleason score (GSC) or by the later introduced International Society of Urological Pathology (ISUP) score [2,3]. In most patients the tumor is still localized and besides local treatment, watch and wait procedures are applied [4]. Nevertheless, PCa ranges from indolent tumors to aggressive metastatic behavior with low survival, making PCa a heterogeneous disease due to the difficulty of risk stratification with current diagnostic tools [5].

Though multiple genomic and transcriptomic-based analyses have been conducted, the results have not yet contributed to an improvement of diagnostics and therapy of PCa patients [6,7,8,9,10]. Further, these alterations observed at the genetic and transcriptomic level have not been translated to changes at the protein level. Recent studies have shown that aggressive prostate tumors exhibit metabolic changes [6,7,8,9], which highlights the need for a better understanding of PCa metabolism. The prostate undergoes several metabolic shifts during progression from benign neoplastic tissue to aggressive PCa [11,12,13,14]. Despite known alterations in glucose and lipid metabolism and resistance towards androgen deprivation [15], oxidative phosphorylation (OXPHOS) plays a major role in aggressive PCa development [11]. This highlights the necessity to identify significant correlations between genomic/transcriptomic data and protein expression as well as patient outcomes. In an earlier study, our group showed aggressive metastatic tumor growth in mice after deletion of signal transducer and activator of transcription 3 (STAT3) and phosphatase and tensin homolog (PTEN) in a murine transgenic mouse model for prostate cancer [16]. Gene co-expression network analysis of transcriptomic data from human PCa identified upregulation of OXPHOS, which inversely correlated to STAT3 expression.

Subsequent analyses of laser-microdissection-derived proteomes of human and murine prostate formalin-fixed paraffin-embedded tissue (FFPE) samples confirmed enhanced tricarboxylic acid cycle (TCA)/OXPHOS pathway proteins, which led to the identification of reduced pyruvate dehydrogenase kinase 4 (*PDK4*) gene expression, which is an essential regulator of TCA, as a promising independent prognostic risk-marker for PCa [17,18].

In this study, we emphasize the observed proteomic shift of mitochondrial metabolism and specifically of OXPHOS. Other studies have claimed a metabolic shift towards OXPHOS in PCa cell lines in vitro and by patient data analyses, but without showing a clear connection between transcriptomic alterations and associated protein levels [19,20,21,22]. Here, we confirm a metabolic shift towards OXPHOS in aggressive PCa and show in particular upregulation of NDUFS1 and ATP5O, which are part of this process. Enhanced expression of these two proteins was determined by immunohistochemistry (IHC) in a human PCa patient cohort and was found significantly associated with higher risk of biochemical recurrence (BCR), most likely independent of the GSC or ISUP score. Moreover, analysis of independent PCa patient cohort data revealed a concordant increase of NDUFS1/ATP5O at mRNA and protein level. Our results suggest that NDUFS1 and ATP5O are novel prognostic markers for aggressive PCa with poor clinical outcome.

## 2. Methods

Any sample and data acquisition were conducted anonymized, retrospectively and without the possibility of connection to the individual patient.

### 2.1. Formalin-Fixed Clinical Specimens

Formalin-fixed and paraffin-embedded (FFPE) prostate material was acquired from the Department of Pathology of the Medical University of Vienna, Austria. The specimens consisted of material from 88 patients with primary PCa and 7 patients with bladder cancer who underwent radical prostatectomy at the General Hospital of Vienna from 1993 to 2015.

### 2.2. Human Tissue Microarray Generation: Sample Selection and Preparation for Laser Microdissection

Human tissue-microarray (TMA) generation as well as sample selection and preparation for laser microdissection were conducted as described in our previous study by Oberhuber et al. [17]: For generation of a TMA, we used the above mentioned FFPE material, which consists of tumor as well as adjacent benign prostate areas. On the created slides, the respective areas were marked by an experienced pathologist (L.K.). For the TMA, cores of 2 mm diameter were cut out of the donor block and positioned into the TMA block.

### 2.3. Proteomic Liquid Chromatography Tandem Mass Spectrometry (LC-MS/MS) Analysis

Proteomic analyses were conducted as described in our previous study [17] and data analysis was performed as following. Data processing was conducted with the Perseus software v.1.5.8.6 [23]. Identified protein groups were first filtered by removing proteins only identified by site, reverse hits, and potential contaminants. After log2 transformation of label-free quantification (LFQ) intensities, biological replicates were grouped. Intensities were then filtered for a minimum of 70% valid values per group, after which missing data points were replaced by imputation. Principal component analysis (PCA) and unsupervised hierarchical clustering were used to remove outliers in each group. ANOVA multi-sample test (permutation-based 5% FDR, 250 randomizations) was performed on the resulting dataset, and the significant differentially expressed proteins were grouped by unsupervised hierarchical clustering.

### 2.4. Immunohistochemistry

IHC was conducted on FFPE TMAs using consecutive sections. The following antibodies were used: anti-NDUFS1 antibody (rabbit polyclonal, 1:100 dilution; abcam; ab169540) and anti-ATP5O antibody (rabbit polyclonal, 1:200 dilution; proteintech, 10994-1-AP). Staining was performed using the BenchMark ULTRA automated staining system (Ventana Medical Systems, Tucson, Arizona, US, now part of Roche, Basel, Switzerland) as following. After deparaffinization and heat pretreatment, antigen retrieval with CC1 buffer (pH 6) for 64 min was performed, primary antibodies were incubated for 32 min and counterstaining with hematoxylin and bluing reagent for 8 min each was conducted. After automated staining, the slides were washed with water, then dehydrated in increasing concentrations of ethanol (70%, 80%, 96%, absolute alcohol) until xylol, covered with the mounting medium Shandon Consul-Mount^TM^ (Thermo Fisher Scientific, Waltham, MA, US) and analyzed by standard light microscopy. Antibodies were validated for FFPE IHC. Appropriate positive and negative control stainings were conducted. The samples were analyzed using an Olympus BH-2 microscope. The average of the core stains of PCa cells was used to determine the staining intensity by microscopic examination, which scored from negative (0–0.5), low (1–1.5), intermediate (2–2.5), to strong (3), by three independent expert pathologists (R.W., B.T., L.K.).

### 2.5. Oncomine Database Analysis

Gene expression data were extracted from the Oncomine™ Research Premium Edition database (Thermo Fisher Scientific, Ann Arbor, MI, USA) [24]. Cancer vs. normal tissue and hormone response analyses were performed using the Tomlins Prostate dataset for *ATP5O* (reporter: IMAGE:1472150 (1)) and *NDUFS1* (reporters: IMAGE:491435 and IMAGE:753457). Primary vs. metastasis analyses were performed using the Magee Prostate dataset for *ATP5O* (reporter: X83218_at), and the Ramaswamy Multi-cancer dataset for *NDUFS1* (reporter: X61100_rna1_at).

### 2.6. TCGA and GTEx Project Analyses

Transcript per Million (TPM) counts for NDUFS1 (ENSG00000023228) and ATP5O (ENSG00000241837) were downloaded from Xena Functional Genomics Explorer (https://xenabrowser.net, accessed on 20 November 2021) [25]. Xena browser hosts data from the UCSC Toil RNA-seq recompute compendium, a uniformly realigned and re-called gene and transcript expression dataset for all The Cancer Genome Atlas-Prostate Adenocarcinoma (TCGA-PRAD) as well as GTEx (Genotype-Tissue Expression) samples, allowing for comparison of TCGA-PRAD tumor samples and corresponding GTEx normal samples [25,26,27]. Only primary tumor samples from TCGA-PRAD with pathology stages T3a, T3b, or T4 (*n* = 301) based on American Joint Committee on Cancer (AJCC) staging criteria were selected for further analysis. Primary tumor samples were then compared to both adjacent normal tissue samples (*n* = 52) from TCGA-PRAD and true normal tissue samples (cancer-free patients) from GTEx (*n* = 100).

### 2.7. Statistical Analysis

Statistical analysis was performed with Prism version 9 software (GraphPad Software, San Diego, CA, USA) and R (v4.0.3) using the Wilcoxon rank-sum test. For all analyses, a *p* value of <0.05 was considered statistically significant.

## 3. Results 

### 3.1. Human Proteomics Data Analysis Shows Upregulation of Mitochondrial Processes and OXPHOS in STAT3-Low PCa

In a recent study applying shotgun proteomic analysis of human PCa samples [17], several mitochondrial processes and OXPHOS were found upregulated in highly aggressive human STAT3-low PCa samples. However, the heterogeneity within the different patient groups made it difficult to obtain statistically significant information on differentially expressed proteins.

Therefore, in this study we specifically focused on homogeneous STAT3-low and STAT3-high patient subgroups chosen from a tissue microarray (TMA) after IHC staining of STAT3. Control samples were obtained from patients following bladder cancer prostatectomy that contained healthy prostate tissue. After evaluating the consistency of biological replicates in the proteomics data using principal component analysis and unsupervised hierarchical clustering, we removed samples with poor coverage and outliers. Thus, we selected the four most homogenous samples per group (healthy prostate control, STAT3-low, STAT3-high) to perform stringent statistical analysis.

A total of 153 differentially expressed proteins were identified across all groups (ANOVA *p =* 0.0063 - <0.0001), out of which 46 proteins were associated with mitochondrial processes and/or OXPHOS in particular (Appendix A). When comparing the STAT3-low and STAT3-high group, only one protein was downregulated, whereas 45 proteins were upregulated (Appendix A), which emphasized a shift towards increased mitochondrial activity, especially upregulation of the OXPHOS pathway. This pathway consists of five different protein complexes (I–V) located in the inner mitochondrial membrane for production of adenosine triphosphate (ATP). At least one protein from each complex of the OXPHOS pathway was found upregulated, comparing the STAT3-low and STAT3-high group (Appendix A, highlighted and Figure 1).

Within Complex I, an enzyme complex of 46 sub-components, NADH-coenzyme Q oxidoreductase NADH-ubiquinone oxidoreductase 75 kDa subunit (NDUFS1) was identified (fold change 5.22, *p* = 0.0027). This protein is located on the matrix-facing side of complex I and converts the reduced form of nicotinamide adenine dinucleotide (NADH) to its oxidized form (NAD^+^) [28]. Succinate dehydrogenase iron-sulfur subunit (SDHB) was detected as part of Complex II (fold change 6.45, *p* = 0.0007). In a complex with three other proteins, SDHB catalyzes the oxidation of succinate to fumarate [29].

Within Complex III, Cytochrome b-c1 complex subunit 8 (UQCRQ) was found upregulated (fold change 5.998, *p* = 0.0013). This complex plays a major role in OXPHOS as an electron-transferring protein that uses heme groups as redox substrates [30]. Cytochrome c oxidase subunit 2 (MT-CO2) was detected as part of Complex IV (fold change 4.456, *p* = 0.0012). It transfers protons across the inner membrane of the mitochondria to the inter-membrane space, while transferring electrons to oxygen and hydrogen [31]. Within Complex V, ATP5A1 (fold change 6.848, *p* = 0.0004), ATPC1 (fold change 4.515, *p* = 0.0027) and ATP5O (fold change 5.02, *p* = 0.004) were identified. ATP5A1 is part of the subunit alpha, ATP5C1 is part of the subunit gamma and ATP5O is part of the subunit O. As part of complex V, they catalyze ATP synthesis when protons flow back into the mitochondrial matrix [32].

### 3.2. NDUFS1 and ATP5O Protein Expression in PCa 

Based on the proteomic results described above, we conducted a literature search regarding the differentially expressed mitochondrial proteins (Figure 1, Appendix A). We did not find previous data regarding NDUFS1 and ATP5O expression in PCa. Thus, we selected NDUFS1 from Complex I and ATP5O from Complex V for further validation and evaluation of PCa clinical data.

We conducted IHC stainings of NDUFS1 and ATP5O in PCa and adjacent tumor-free tissue in a cohort of 88 patients who underwent radical prostatectomy. Detailed patient characteristics are described in Table 1. Due to missing data, we had to exclude 14 patients (NDUFS1) and 13 patients (ATP5O) (Table 1). An intensity scale from zero to three was introduced, which was further subdivided into 0.5 values. We then assigned patients to two groups, with either low (0–1.5) or high expression (2–3) of the proteins (Figure 2). Only one tissue specimen showed negative staining for both proteins (not shown) in PCa. Both NDUFS1 and ATP5O showed enhanced expression in PCa compared to adjacent benign tissue (paired samples *t*-test for NDUFS1 *p* < 0.0001 and for ATP5O *p* < 0.0002, Figure 3A). High NDUFS1 expression was observed in 67.6% of the samples and in 86.8% for ATP5O (Figure 3B). Further, NDUFS1 and ATP5O expression were highly correlated (Pearson correlation coefficient 0.576, *p* = 0.01), confirming the proteomic data and suggesting that these components of Complex I and V are upregulated concertedly.

### 3.3. NDUFS1 and ATP5O Expression Is Associated with Earlier Biochemical Recurrence in PCa

We next conducted Kaplan Meier survival analysis with the same IHC study population of 88 PCa patients to analyze the correlation of NDUFS1 and ATP5O expression with biochemical recurrence (BCR). Due to missing clinical data we had to exclude 20 patients for GSC, 19 patients for ISUP, 27 patients for NDUFS1, and 26 patients for ATP5O analysis. To test whether the diagnostic GSC/ISUP scheme correlated with expected time to BCR, we conducted survival analysis taking into account the GSC and ISUP scores. Both higher GSC and higher ISUP scores were significantly associated with earlier time to BCR (Appendix B
Figure A1A,B). As stated before, patients were assigned to low and high expression groups. High NDUFS1 expression was significantly *(p* = 0.0294) associated with earlier BCR (Figure 4A), and a similar result was observed for high expression of ATP5O (*p* = 0.0151) (Figure 4B), suggesting that enhanced expression of NDUFS1 and ATP5O correlate with highly aggressive PCa. 

### 3.4. mRNA Analysis Shows Higher Expression of NDUFS1 and ATP5O in PCa

We aimed to examine whether *NDUFS1* and *ATP5O* mRNA expression levels were changing correspondingly in independent PCa patient datasets. First, gene expression analysis was performed using datasets from the Oncomine™ platform (Figure 5A,B). Transcript levels of both, *NDUFS1* and *ATP5O*, were significantly enhanced in intraepithelial neoplasia and PCa compared to benign prostate glands (*p* < 0.05). These findings indicated concurrent changes of *NDUFS1* and *ATP5O* at the transcriptomic level, which also manifested at the protein level. Additional findings from analyses of the Oncomine™ platform showed significantly elevated transcript levels of both genes in metastases compared to the primary site in PCa patient samples (Appendix B
Figure A2A,B). Additionally, in hormone refractory PCa compared to hormone naïve samples, significantly higher *NDUFS1* and *ATP5O* mRNA levels were detected (Appendix B
Figure A3A,B).

To extend the analysis with data from higher patient numbers, further gene expression analyses were performed using The Cancer Genome Atlas-Prostate Adenocarcinoma (TCGA-PRAD) data collection and the Genotype-Tissue Expression (GTEx) project data. Comparison of mRNA expression in tumors versus adjacent benign tissue from the TCGA-PRAD data showed no significant difference (Appendix B
Figure A4A,B). However, when comparing mRNA levels in PCa tissue from the TCGA-PRAD data versus healthy tissue extracted from the GTEx data, NDUFS1 and ATP5O transcript levels were significantly enhanced in PCa (*p* < 0.05), as shown in Figure 5C, D, underscoring the proteomic findings.

## 4. Discussion

Our study, using proteomic and transcriptomic data analysis, is the first to reveal a concordant increase of NDUFS1/ATP5O in mRNA expression in transcriptomic datasets and at protein level in aggressive PCa, which introduces two novel prognostic markers,—NDUFS1 and ATP5O. In a recent study from our group, Oberhuber et al. showed upregulation of TCA and OXPHOS in PCa at transcriptomic and protein level, which inversely correlated with STAT3 expression [17]. In our current study, using homogeneous subgroups of STAT3-low versus STAT3-high samples, we identified 153 differentially expressed proteins compared to the healthy control group. A total of 45 proteins specifically associated with mitochondrial processes and OXPHOS were upregulated, underlining the importance of OXPHOS in PCa development and progression. It is known that mitochondrial processes in general and OXHPOS in particular are important drivers for cancer progression [33,34]. Iglesias-Gato et al. revealed elevated OXPHOS capacity in PCa [35], which correlates with our findings. In the STAT3-low cohort we identified at least one significantly upregulated protein within each of the five OXPHOS complexes. In subsequent IHC analysis, two proteins that are components of Complex I (NDUFS1) and Complex V (ATP5O) were confirmed to have higher expression in PCa tissue compared to matched benign regions. To this date, analyses of these two proteins were not yet described in PCa. Differences in NDUFS1 and ATP5O expression levels have only been described in clear cell renal carcinoma [36,37] and lung cancer patient samples [38] so far. The increased expression as well as interdependent correlation of NDUFS1 and ATP5O IHC expression levels suggest concomitant upregulation of Complex I and Complex V components. The findings additionally underline enhanced activity of OXPHOS in aggressive PCa and indicate the important role of Complex I and V in mitochondrial energy production. Mitochondrial dysfunction in cancer has been recognized [39], and increased NDUFS1 and ATP5O expression has also been described as unfavorable in breast [40] and gastric cancer [34]. Importantly, here we show that NDUFS1 and ATP5O are significant predictors of earlier BCR in PCa, making these proteins strong candidates as novel biomarkers.

Taken together, the findings indicate that NDUFS1/ATP5O could serve for risk stratification in PCa. In advanced or high-risk localized PCa, under-treatment remains a challenge regarding treatment options [41]. We anticipate that NDUFS1/ATP5O may serve as additional markers to identify high-risk patients, who cannot be stratified by the Gleason or ISUP score. We show here that proteomic and IHC results can be linked with the transcriptomic level, suggesting that interrelated shifts in OXPHOS are occurring at the transcriptomic and protein level in PCa. The analyses of the Oncomine^TM^ dataset showed a significant difference of transcript levels (Figure 5A,B), as well as the analysis from the TCGA-PRAD data when using PCa samples compared to healthy tissue samples from the GTEx project (Figure 5C,D). Interestingly we could not show this effect when comparing PCa with benign tissue in the TCGA-PRAD samples (Figure A4A,B). Benign samples from the TCGA-PRAD data contain only adjacent tissue from patients with PCa, the benign samples in the GTEx project data consist of tissue acquired from healthy individuals, while the benign samples in the Oncomine^TM^ platform are a combination of both, adjacent benign and healthy tissue. These results raise the question of how much the benign tissues in the TCGA-PRAD program can be considered as healthy controls, or if it is already subject to the influence of the different cell types within the tumor-microenvironment. It has been suggested that this type of tissue should be regarded as an intermediate form between healthy and tumor [42]. Furthermore, it should be noted that protein abundances show a larger dynamic range than mRNA levels, highlighting the importance of post-transcriptional regulators, which has been observed recently in a comprehensive multi-omics analysis of prostate cancer [43]. Such a correlation of mRNA and protein expression of similar genes for mitochondrial processes such as OXPHOS, based on the analysis of human PCa tissue samples, has not yet been demonstrated in other studies [19,20,21,22,42].

In addition, higher expression of these two proteins in hormone refractory samples—often as a result of more advanced or progressed PCa [43]—highlights the role of enhanced activity of OXPHOS for PCa dissemination. The increased mRNA expression in PCa metastases underscores our association of NDUFS1 and ATP5O with lower survival and more aggressive disease outcomes, as described above.

PCa undergoes metabolic reprogramming towards OXPHOS to meet its energy needs, which contradicts the Warburg theory. In particular, drug-resistant, oncogene-dependent tumors rely on OXPHOS as a survival mechanism. Several signaling pathways converge on STAT3, which is why STAT3 may be a link between cellular signaling pathways and cancer cell metabolism. One way to treat therapy-resistant, oncogene-dependent PCa would be to develop specific OXPHOS inhibitors as well as biomarkers for OXPHOS inhibition. Our data support NDUFS1 and ATP5O as possible biomarkers for high OXPHOS in aggressive PCa.

Possible limitations of our study are due to the retrospective design of the latter and small group sizes regarding hormone refractory and metastatic samples. Therefore, further prospective studies are needed to reliably integrate the proteins we identified, NDUFS1 and ATP5O, into future risk stratification models. Within such a study, the relationship between transcriptomic data and protein levels can also be validated.

## 5. Conclusions

We show here for the first time a concordant increase of NDUFS1/ATP5O at protein and mRNA expression levels in human PCa samples. Our study provides evidence for two new prognostic biomarkers for risk stratification in PCa. 

## Figures and Tables

**Figure 1 cancers-13-06036-f001:**
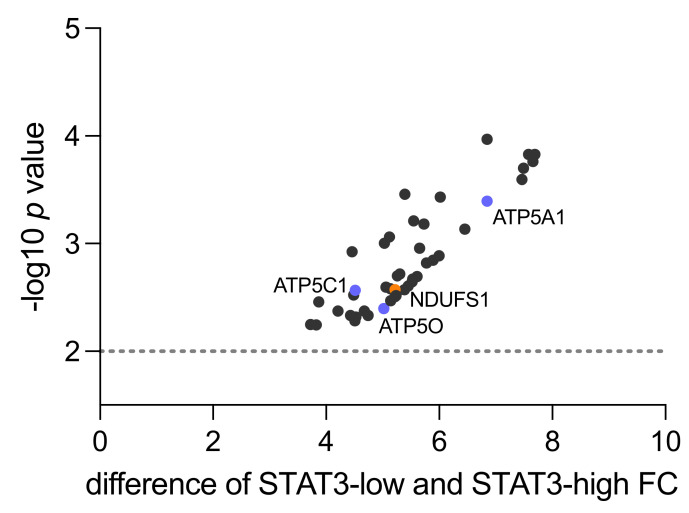
Mitochondrial proteins upregulated in STAT3-low versus STAT3-high samples in proteomic shotgun analysis (*n* = 4). The y-axis shows the *p* value as −log10 and the x-axis indicates the difference in fold change (FC) between low vs. high STAT3 samples. Highlighted are NDUFS1 from complex I (orange) and ATP5O, ATP5A1 and ATP5C1 from complex V (blue). The dotted line represents level of 0.01 significance.

**Figure 2 cancers-13-06036-f002:**
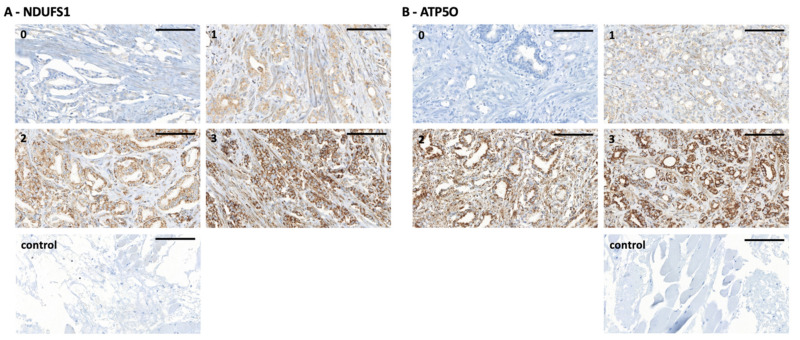
(**A**) NDUFS1 immunohistochemistry staining of PCa tissue microarray (TMA) samples. (0) staining intensity of 0 to 0.5, (1) staining intensity of 1, (2) staining intensity of 2, (3) staining intensity of 3, (control) technical negative control staining (left). Scale bar = 100 µm. (**B**) ATP5O immunohistochemistry staining of PCa TMA samples. (0) staining intensity of 0 to 0.5, (1) staining intensity of 1, (2) staining intensity of 2, (3) staining intensity of 3, (control) technical negative control staining (right). Scale bar = 100 µm.

**Figure 3 cancers-13-06036-f003:**
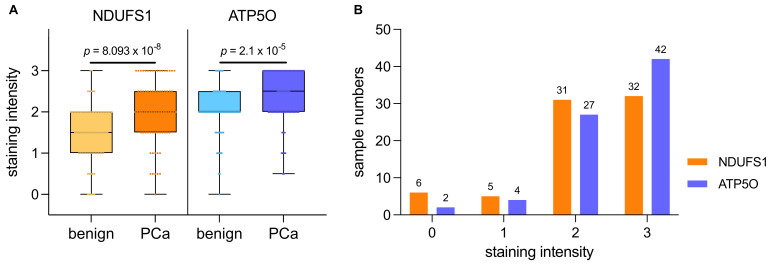
(**A**) IHC staining intensities for NDUFS1 and ATP5O ranging from 0 to 3 in PCa samples and adjacent benign tissue. Boxplots show the interquartile range with horizontal line as median, whiskers as upper and lower limits. (**B**) Histogram of staining intensities for NDUFS1 and ATP5O ranging from 0 to 3 in PCa samples. Value 0 includes staining of 0 and 0.5, value 1 includes staining of 1 and 1.5, and value 2 includes staining of 2 and 2.5.

**Figure 4 cancers-13-06036-f004:**
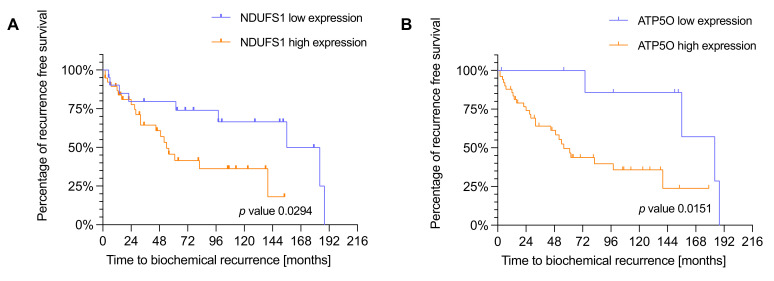
(**A**) Kaplan-Meier plot showing time to biochemical recurrence in months for NDUFS1 in the IHC cohort (*n* = 60). *p*-values were estimated by log-rank test. Hazard ratio of 2.188 with a confidence interval of 1.056 to 4.530. Blue = low expression, orange = high expression. (**B**) Kaplan-Meier plot showing time to biochemical recurrence in months for ATP5O in the IHC cohort (*n* = 61). *p*-values were estimated by log-rank test. Hazard ratio of 2.953 with a confidence interval of 1.364 to 6.391. Blue = low expression, orange = high expression.

**Figure 5 cancers-13-06036-f005:**
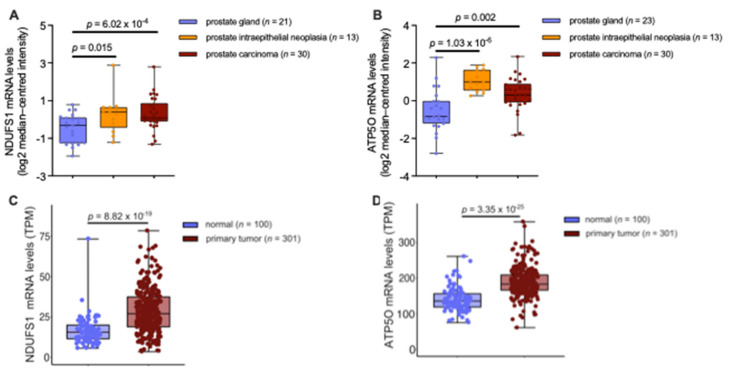
(**A**) *NDUFS1* mRNA expression levels analyzed from prostate carcinoma epithelia (*n* = 30; 1.74-fold increase), prostatic intraepithelial neoplasia epithelia (*n* = 13; 1.69-fold increase), or normal prostate gland samples (*n* = 21). (**B**) *ATP5O* mRNA expression levels analyzed from prostate carcinoma epithelia (*n* = 30; 1.83-fold increase), prostatic intraepithelial neoplasia epithelia (*n* = 13; 3.01-fold increase), or normal prostate gland samples (*n* = 23). (**C**) NDUFS1 mRNA expression levels (TPM) analyzed from TCGA-PRAD primary tumor tissue (*n* = 301) and true normal tissue samples (cancer-free patients) from GTEx (*n* = 100). (**D**) ATP5O mRNA expression levels (TPM) analyzed from TCGA-PRAD primary tumor tissue (*n* = 301) and true normal tissue samples (cancer-free patients) from GTEx (*n* = 100). Data for (**A,B**) were extracted from the Oncomine™ Platform from the Tomlins Prostate dataset. Data for (**C**–**D**) were extracted from The Cancer Genome Atlas-Prostate Adenocarcinoma data collection and the Genotype-Tissue Expression project. Representation: boxes as interquartile range, horizontal line as the mean, whiskers as lower and upper limits.

**Table 1 cancers-13-06036-t001:** Clinical characteristics of the PCa patient cohort stained for NDUFS1 and ATPO.

Characteristic	Total Cohort	CI
Number of patients	88	
Valid values for NDUFS1	74	
Valid values for ATP5O	75	
Mean age at surgery	63.3	6.42
Gleason score, *n* (%)	87	
3	5 (5.7)	
4	6 (6.9)	
5	10 (11.5)	
6	12 (13.8)	
7	34 (39)	
8	15 (17.2)	
9	5 (5.7)	
ISUP, *n* (%)	87	
1	33 (37.9)	
2	22 (25.3)	
3	12 (13.8)	
4	15 (17.2)	
5	5 (5.7)	
Clinical T stage, *n* (%)		
T3a	84 (95.5)	
T3b	4 (4.5)	

CI—confidence interval.

## Data Availability

Proteomics data are available at https://www.ebi.ac.uk/pride/archive/projects/PXD014251. The results from TCGA-PRAD shown here are in whole or part based upon data generated by the TCGA-PRAD Research Network: https://www.cancer.gov/tcga.

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
