# Peer review of "Proteomic Analysis Identifies NDUFS1 and ATP5O as Novel Markers for Survival Outcome in Prostate Cancer"

_cancers, 2021, doi:10.3390/cancers13236036_

Round 1

Reviewer 1 Report

In the article titled “Proteomic analysis identifies NDUFS1 and ATP5O as novel markers for survival outcome in prostate cancer” authors use a multiprong approach to identify two new potential gene markers of the prostate cancers. Authors use preserved patient samples and perform a proteomic analysis. They number of differentially expressed proteins and focus on two that haven’t been described before. They show expression of NDUFS1 and ATP5O by staining and retrospective survival analysis and mRNA analysis. The article is written in very clear and easy to follow manner and with appropriate level of careful interpretation. While most of the results are solid, the mRNA analysis is suffering number of weaknesses that should be addressed before publication. I recommend that authors represent the expression of mRNA in one of the commonly published units (see below) and they should show the variability in expression by depicting points overlayed in boxplots. I would like to caution authors against making strong interpretation when comparing expression of genes that are at low end of normal expression values. For example, 0.1 TPM vs 1 TPM leads 10-fold statistical difference, however, due to limitations in the sequencing technologies these may better translate to OFF and ON signal, or not mean any expression if one takes common expression cutoff of 1 TPM. I suggest authors consider alternatives to Oncomine, such as TCGA, that may have more normalized data for their genes of interest.

Comments and Suggestions:

Line 40: BCR acronym needs to be explained before it is introduced

Line 102: Please provide brief description of the method

Line 107: What are LFQ intensities?

Line 139: Please provide analysis of secondary data such as TCGA

Tables: Please provide the description of the units in your spreadsheets (i.e. [log2FoldChange]) as well as the comparison performed in STAT3-high and STAT3-low columns.

Line 170: Does the value 5.28 come from tables S1 or S2, because that exact value was not present there. The value in tables is 5.22. Please correct or provide updated tables.

Line 312: Please tissue microarray instead of TMA for simplicity.

Figure 2: Please describe the method used to evaluate the staining intensity. Was this performed by automated image and pixel intensity analysis or visual inspection by a researcher? This section is missing in methods.

Figure 3A: Please show all the points in an overlay with the boxplot to show the patient distribution / variability. The legend states these are staining intensities, however figure axis says expression values. Please clarify.  

Line 228: Remind reader of the meaning of BCR.

Figure 5: The expression of mRNA are commonly expressed in transcripts per million (TPM), reads per kilobase million (RPKM) or normalized counts. Please plot mRNA expression of the proteins in any of the commonly used units. I am not sure what log2 median-centered intensity means, but it is certainly not a standard measure of mRNA expression. If possible, please overlay individual data points over the boxplots to show the variability. If Oncomine does not provide values in these common units, please visit publicly available www.cancer.gov (TCGA) where this data certainly exists.

Best regards,

Author Response

We thank the reviewer for the very important and constructive suggestions and comments that we have amended and marked in the revised version of the manuscript and the responses can be found in the word document attached.    Kind regards,
Robert Wiebringhaus

Reviewer 2 Report

The manuscript entitled “Proteomic analysis identifies NDUFS1 and ATP5O as novel markers for survival outcome in prostate cancer” by Robert Wiebringhaus and colleagues found increased mRNA expression levels of NDUFS1/ATP5O in intra-epithelial neoplasia and in PCa compared to benign prostate glands. Moreover, increased NDUFS1/ATP5O protein and mRNA expression were found in  aggressive PCa suggesting a potential role of NDUFS1/ATP5O in identifying high risk PCa patients.

The comments to this manuscript have been reported below.

  • Figure 2 should include the images of positive and negative controls for NDUFS1 and ATP5O
  • the authors found increased mRNA levels of both NDUFS1 and ATP5O in intraepithelial neoplasia (PIN) and PCa compared to benign prostate glands. Due to the post-transcriptional regulation of mRNA (e.g.  by miRNA) it would be useful to know the expression of NDUFS1 and ATP5O at protein levels (e.g. by IHC) in PIN and benign prostate.
  • Did the authors evaluate a possible different expression of NDUFS1 and ATP5O according to the Gleason score?
  • lines 300-304: the authors found increased NDUFS1 and ATP5O mRNA levels in metastases compared to the primary PCa and in hormone refractory PCa compared to hormone naïve samples. However, they analysed only 4 metastases and 3 hormone naïve samples, a very small sample size to reach these conclusions.

Author Response

We thank the reviewer for the helpful comments and feedback which we have addressed in the new version of the manuscript and the responses can be found in the word document attached.    Kind regards,
Robert Wiebringhaus
